# Pilot cluster randomised trial of an evidence-based intervention to reduce avoidable hospital admissions in nursing home residents (Better Health in Residents of Care Homes with Nursing—BHiRCH-NH Study)

Elizabeth L Sampson ,[1,2] Alexandra Feast,[1] Alan Blighe,[3] Katherine Froggatt,[4] Rachael Hunter ,[5] Louise Marston,[5] Brendan McCormack,[6] Shirley Nurock,[1] Monica Panca,[5] Catherine Powell ,[7] Greta Rait,[5] Louise Robinson ,[8] Barbara Woodward-Carlton,[3] John Young,[9] Murna Downs[3]

► Prepublication history and additional materials for this paper are available online. To view these files, please visit the journal online (http://dx.doi.org/10.1136/bmjopen-2020-040732).

**Correspondence to**
Professor Elizabeth L Sampson;
e.sampson@ucl.ac.uk

## ABSTRACT

**Objectives** To pilot a complex intervention to support healthcare and improve early detection and treatment for common health conditions experienced by nursing home (NH) residents.

**Design** Pilot cluster randomised controlled trial.

**Setting** 14 NHs (7 intervention, 7 control) in London and West Yorkshire.

**Participants** NH residents, their family carers and staff.

**Intervention** Complex intervention to support healthcare and improve early detection and treatment of urinary tract and respiratory infections, chronic heart failure and dehydration, comprising: (1) 'Stop and Watch (S&W)' early warning tool for changes in physical health, (2) condition-specific care pathway and (3) Situation, Background, Assessment and Recommendation tool to enhance communication with primary care. Implementation was supported by Practice Development Champions, a Practice Development Support Group and regular telephone coaching with external facilitators.

**Outcome measures** Data on NH (quality ratings, size, ownership), residents, family carers and staff demographics during the month prior to intervention and subsequently, numbers of admissions, accident and emergency visits, and unscheduled general practitioner visits monthly for 6 months during intervention. We collected data on how the intervention was used, healthcare resource use and quality of life data for economic evaluation. We assessed recruitment and retention, and whether a full trial was warranted.

**Results** We recruited 14 NHs, 148 staff, 95 family carers and 245 residents. We retained the majority of participants recruited (95%). 15% of residents had an unplanned hospital admission for one of the four study conditions. We were able to collect sufficient questionnaire data (all over 96% complete). No NH implemented intervention tools as planned. Only 16 S&W forms and 8 care pathways were completed. There was no evidence of harm.

## Strengths and limitations of this study

► The intervention was adapted from a successful US programme, co-designed with residents, family carers and staff to fit with the UK nursing home context.

► Successful recruitment and retention of nursing homes, their staff, residents and family carers demonstrated the feasibility of our study methods.

► It was challenging to collect reliable data on hospital admissions, ambulance and general practitioner visits.

► The nature of the intervention in this cluster randomised trial meant that outcome data were not collected blind to allocation for intervention or control group.

**Conclusions** Recruitment, retention and data collection processes were effective but the intervention not implemented. A full trial is not warranted.

**Trial registration number** ISRCTN74109734 (https://doi.org/10.1186/ISRCTN74109734).

**Original protocol** *BMJ Open*. 2019;9(5):e026510. doi:10.1136/bmjopen-2018-026510.

## INTRODUCTION

Currently in the UK more than 420 000 people aged over 65 years live in residential care, of which approximately 220 000 reside in care homes with nursing (referred to in this paper as 'nursing homes' (NHs)).[1] Most UK NHs are owned by private companies and residents pay on a means-tested basis. Unlike some European NHs, for example in the Netherlands, there is no on-site provision of medical care. NH residents are served by a general practitioner and other visiting staff

such as specialist nurses. NH residents have complex healthcare needs with high levels of multimorbidity, frailty and dementia. The King's Fund[2] and British Geriatrics Society[3] have raised concerns about the inconsistency and quality of healthcare provision to NHs.

There has been a 63% increase in all-cause hospital admissions from NHs between 2011 and 2015.[4] As well as causing distress to residents, their families and staff, hospitalisation is expensive for health and social care systems, costing an estimated £1.2 billion per-annum in the UK.[5] Hospital admission increases risk of decline in functional ability, delirium, adverse events and prolonged stays.[6 7]

Ambulatory Care Sensitive Conditions (ACSCs) are 'conditions that can lead to unplanned hospital admissions that may have been avoidable or manageable by timely access to medical care in the community'.[8 9] Four ACSCs contribute to a 30% of hospitalisations from NHs[5]: respiratory infections[8 10–12]; acute exacerbation of chronic heart failure (CHF)[13 14]; urinary tract infections (UTIs)[8 15] and dehydration.[8]

A number of interventions have been developed to enhance healthcare in NHs, falling broadly into two categories: single component interventions (predominantly advanced care planning or single-disease care pathways, eg, for pneumonia) and multicomponent interventions which include enhancing staff knowledge and skills,[16] clinical guidance and decision-support tools (care pathways), engaging with families,[17] and input from geriatricians or nurse practitioners.[18] Research highlights the importance of collaborative intervention development with staff,[19] residents and families,[17] and using local champions to support implementation.[14] 'INTERACT' (Interventions to Reduce Acute Care Transfers), developed in the USA, focuses on managing acute changes in residents' condition and reduces transfers to acute hospitals.[14] We worked with stakeholders and our family Carer Reference Panel (CRP) to develop and adapt INTERACT for use in the UK.[20] This included 18 semistructured interviews and three co-design workshops over 5 months, with amendments made to the intervention after each workshop. Participants comprised 22 diverse stakeholders (two NH managers, three care assistants, eight nurses, four general practitioners, three family carers, a geriatrician and a quality improvement manager) (paper in preparation). The Promoting Action on Research Implementation in Health Services (PARiHS) framework underpinned co-design of implementation support and guidance.[21]

## Aim and objectives

Our aim was to indicate whether a definitive study of the Better Health in Residents of Care Homes with Nursing (BHiRCH-NH) intervention is warranted.

## Primary objective

To indicate whether the intervention was acceptable and feasible.

## Secondary objectives

1. Establish whether consent procedures facilitate collection of sufficient individual-level data.
2. Assess intervention fidelity.
3. Assess the effectiveness of the implementation strategy and level of staff engagement with the intervention.
4. Indicate whether the intervention would be sustainable outside the trial context.
5. Assess potential primary and secondary outcome measures for a definitive trial.
6. Measure completeness of data collection, documentation, return rate of questionnaires, and assess potential primary and secondary outcomes for a definitive trial.
7. Assess feasibility of collecting data for economic evaluation.

## METHODS
### Trial design

A pilot cluster randomised trial in NHs, conducted and reported as per the Consolidated Standards of Reporting Trials guidance.[22] Detailed methods are described in the trial protocol paper.[20] The NH was the unit of allocation (seven intervention and seven control sites).

### Patient and public involvement

The project was developed in collaboration with the UK Dementia and Neurodegenerative Diseases Network. Patient and public involvement representatives (SN and BW-C) were grant co-applicants. Two CRPs ensured public involvement at all stages. Each comprised eight family carers of people with dementia and a person living with dementia, supported by Alzheimer's Society research volunteer network. They collaborated on intervention design, recruitment and consent processes, accessibility of information leaflets, data collection, interpretation and dissemination.

### Study population and eligibility criteria
#### Nursing homes

We recruited 14 NHs (eight in West Yorkshire and six in London) with adequate staffing to implement the intervention and support research. These were identified via local Clinical Research Networks, and the Enabling Research in Care Homes Network, purposively selected to include a range of providers (large and small chains, independent providers), urban, suburban and rural. Managers, regional managers or owners gave written permission. The intervention was an enhanced version of 'usual care' implemented at NH level. Therefore, individual consent was not required and all staff were involved in delivering the intervention to all NH residents.

#### Individual participants

We invited all English-speaking staff and residents over 65 years and their carers (family members or friends) to participate in individual-level data collection until we recruited approximately 20 residents and 10 staff from

each NH. We excluded residents receiving end-of-life care or those who did not wish to be involved in research.

## Consent procedures
### Residents
The NH manager or deputy manager identified all potentially eligible residents. If necessary, we conducted a capacity assessment regarding trial participation, adhering to the UK Mental Capacity Act (2005). If the resident lacked capacity, we used a personal consultee (friend or family), or if not available, a professional consultee (health or social care staff with a professional relationship to the resident but no connection with the project). If a resident lost capacity during the study, a consultee was found.

### Family carers and staff
We invited a family carer, and NH staff associated with residents recruited to the study, to answer questionnaires and they gave informed consent for this.

## INTERVENTION
The pilot trial ran for 10 months (November 2017–August 2018). Sites were set up in months 1 and 2, in month 3 we collected pre-intervention (baseline) data over 4 weeks before the intervention was implemented. The intervention ran for 6 months with final data collection and site closure in month 10. We planned for the trial to run for 16 months from November 2017 but had to reduce follow-up due to delays in obtaining ethical approvals. Thus, timing of data collection differs from our protocol.[20]

### Implementation support
This was developed consistent with the PARiHS framework to ensure implementation matched individual contexts. It was important for Practice Development Champions (PDCs) to decide on how they approached this, given the philosophy of quality collaboratives and 'champions' in place. This enhanced the role of PDCs in developing their own approach.

### Workshop
The research team delivered a 1-day workshop to two PDCs from each NH comprising an introduction to four key ACSCs (respiratory and UTIs, dehydration, acute exacerbation of CHF) and elements of how to bring about organisational change. We gave an overview of intervention materials.

### Introductory meeting
Researchers held a project initiation meeting with each NH manager and available staff to highlight key intervention components, how staff should deliver the intervention and how this may be integrated into existing local systems. The PARiHS framework supported PDCs and their colleagues to disseminate this information to the wider NH staff.

### Ongoing implementation support
1. We supported PDCs with a project handbook including information on approaches to change used in this project; intervention implementation within differing contexts; tips to help teams learn and act alongside the people for whom they care; and information on enhanced leadership capabilities. PDCs were also guided through the *Practice Development Workbook for Nursing, Health and Social Care Teams: Resources for Health and Social Care Teams.*[23]
2. We expected PDCs to establish a Practice Development Support Group (PDSG) to support their work in the NH. This 'quality collaborative' approach involves diverse stakeholders working together to close the gap between actual and potential practice.[24]
3. The programme manager made weekly contact with the PDCs to collect information on how the intervention was working and was available for advice 'as required'. PDCs were offered monthly telephone coaching by senior nurse researchers on our team (BM and KF).

### The intervention (BHiRCH-NH)
This consisted of three key components, adapted from the INTERACT programme[25] and paper-based as UK NHs have variable use of electronic records:
1. *Stop and Watch early warning tool*: care assistants or nurses used this when they noted a change in a resident's condition. They circled observed changes, notified the nurse and placed the tool in the resident's NH records.
2. *Care pathway*: this was a two-step clinical guidance and decision-support system, focusing on symptoms and signs of four key ACSC conditions (acute exacerbation of CHF, respiratory and UTIs, dehydration). The initial 'primary' assessment comprised screening questions with the potential to trigger a more detailed 'secondary' assessment. If the primary or secondary assessment result was ambiguous, the care pathway was administered at 6-hour intervals, until concerns had resolved and/or appropriate intervention was instigated. The nurse recorded the outcome of the primary and secondary assessment and their care plan in the resident's records and decided on the next course of action. This may have included further monitoring using the Stop and Watch early warning tool, treatment initiated in the NH, or communication with primary care using the Situation, Background, Assessment, Recommendation (SBAR) tool. Copies of the completed care pathway were kept with the resident's record.
3. *The SBAR method:* a structured method for communicating critical information to primary care used by nurses to seek primary care intervention for the resident after the care pathway indicated a risk of decline.

### Treatment as usual NHs
Residents received usual care according to existing local policy and practice. We permitted all medications and treatments.

## Data collection

### Nursing homes

A research facilitator, employed by the NH provided pseudo-anonymised data at the NH level. We collected data for 4 weeks pre-intervention (baseline) including staff turnover and the number of beds available to new residents. For 4 weeks pre-intervention and then monthly for 6 months, we documented the total number of contacts with general practitioners (GPs), ambulances, accident and emergency (A&E) attendances and hospital admissions.

### Residents

We collected data pre-intervention on age, gender, ethnicity, marital status and highest level of education. Functional status was measured using the Barthel Index[26] pre-intervention and at 6 months. We collected data for the month pre-intervention and then for the following 6 months on healthcare use and quality of life (QoL) data for health economic analysis, hospital admission overall and admissions for the four ACSCs of interest, ambulances called, out-of-hours GP visits or telephone contacts, A&E attendances and deaths.

### Family carers

Pre-intervention we collected data on sociodemographics including age, gender, ethnicity and marital status, and preferred role, that is, how involved they would like to be with the resident's medical care.

### Staff

For contextual understanding we documented pre-intervention: staff age, gender, education level and characteristics of their work (qualifications, role, length of employment, shift pattern and first language). Pre-intervention and at 6 months we measured the extent to which they perceived the organisation-supported person-centred care (Person-Centred Care Assessment Tool (P-CAT))[27] and the Nurse Ratings of Communication with Primary Care Questionnaire.[28]

## Outcomes

We collected data in three domains: (1) individual-level data on NH residents, their carers and staff where consent had been obtained, (2) system-level data collected by a research facilitator to provide pseudo-anonymised data at the NH level and (3) process data collected by the study team (table 1).

### Primary objective

To ascertain whether the intervention was acceptable and feasible, we collected data monthly from the NH on intervention use in practice: number of 'Stop and Watch' early warning tools, and primary (initial screening) and secondary (more detailed) assessment tools completed. Where we had pconsent to collect individual-level resident data, we monitored participants monthly for serious adverse events (SAEs) defined as 'any untoward occurrence that resulted in death, was considered life-threatening at the time of the event, required hospitalisation or prolongation of existing hospitalisation, resulted in persistent or significant disability or incapacity, or was any other important medical condition'.

### Secondary objectives

1. Establish whether consent procedures facilitate collection of sufficient individual-level data
   We collected data on consent and recruitment rates of residents, carers and staff.
2. Assess intervention fidelity
   To explore intervention fidelity, two nurse researchers aimed to review a convenience sample of five records for residents admitted to hospital, or received treatment in the NH, for ACSCs. They used a free-text review sheet to record references to trial intervention tools (Stop and Watch, the care pathway and SBAR) and assessed compliance with the care pathways. We noted where NHs made amendments to the structure or content of the care pathway.
3. Assess the effectiveness of the implementation strategy and level of staff engagement with the intervention
   PDCs were offered monthly telephone coaching for advice and to reflect on activities and achievements. They kept an activity log of work with PDSGs to document the level of facilitation required to support implementation. Qualitative interviews were conducted to better understand the barriers and facilitators to implementation. Learning from the implementation strategy will be presented in a separate paper.
4. Indicate whether the intervention would be sustainable outside the trial context
   Data from objectives 2–3 were considered by the independent project steering group at the end of the study.
5. Assess potential primary and secondary outcomes for a definitive trial
   To explore the impact of enhanced healthcare in the NH, we documented hospital admissions for the four key ACSCs (acute exacerbation of CHF, respiratory and UTIs, dehydration). Number (%) of residents requiring one or more ambulance calls, one or more unscheduled out-of-hours GP visits or phone calls, and having one or more A&E department visits were documented as potential secondary outcomes.
   We tested the assumption that a hospitalisation for an ACSC was a proxy for an avoidable hospital admission using the Structured Implicit Record Review (SIRR) tool.[29] Two independent experts (geriatrician and community nurse) used the SIRR, which takes account of the resident's pre-existing health, any advance directives and the care options available at the time to assess the 'avoidability' of the admission.
6. Measure completeness of data collection and documentation, return rate of questionnaires
   We assessed completeness of outcome measures, data collection and return rate of questionnaires.
7. Assess feasibility of collecting data for economic evaluation

**Table 1** Summary of data collected, outcome measures and time schedule

| Data collected and tool used | | Pre- intervention | Monthly | 6 months |
|---|---|---|---|---|
| **Resident** | | | | |
| Sociodemographics | Age, gender, ethnicity, marital status, highest level of education | S | – | – |
| Service use in the prior month | Client Service Receipt Inventory, calculates service and total care costs | S | S | – |
| Functional status | The Barthel Index | S | – | S |
| Resident quality of life (QoL)—self-rated | EQ-5D-5L self-rated health index and Visual Analogue Scale of current health state | P | – | P |
| Resident QoL—proxy rated | EQ-5D—proxy family carer or staff member view of the resident's QoL | FC/S | – | FC/S |
| **Family carer** | | | | |
| Sociodemographics | Age, gender, ethnicity, marital status, years of schooling, highest level of education | FC | – | – |
| QoL | EQ-5D-5L | FC | – | FC |
| Preferred role | How much and how they like to be involved in the residents' care | FC | – | – |
| **Staff** | | | | |
| Staff sociodemographics | Age, gender, ethnicity, number of years of education | R | – | – |
| Staff work characteristics | Highest qualification, role in nursing home, length of service, shift pattern, first language | R | – | – |
| Organisational support for person-centred care | The Person-Centred Care Assessment Tool | S | – | S |
| Communication with primary care | Nurse–GP Communication Needs Assessment Questionnaire | S | – | S |
| Perceived knowledge and skills for early detection in changes in health | Developed from feasibility study; assesses key knowledge and skills needed to implement the intervention; rated on 5-point Likert scale | S | – | S |
| **System-level data** | | | | |
| Number of hospital admissions | Respiratory infection, exacerbation of CHF, UTI and dehydration | S | S | – |
| 'Avoidability' of admissions | Structured Implicit Record Review (Saliba et al, 2000) | S | S | – |
| Use of primary assessment tool | Respiratory infection, exacerbation of CHF, UTI and dehydration | S | S | – |
| Use of secondary assessment | Respiratory infection, exacerbation of CHF, UTI and dehydration | S | S | – |
| Out-of-hours GP contacts | GP visits or telephone contact | S | S | – |
| Ambulances and hospital use | Number of hospital admissions, A&E attendances and readmissions | S | S | – |
| Deaths in the last calendar month | | S | S | – |
| Staff turnover | | S | – | – |
| Nursing home occupancy level | Number of available beds to new residents | S | – | – |

Measure assessed by: P, participant; FC, family carer; R, researcher; S, nursing home staff.
A&E, accident and emergency; CHF, chronic heart failure; GP, general practitioner; UTI, urinary tract infection.

We collected data on resident service use with the Client Service Receipt Inventory (CSRI),[30] completed pre-intervention and then monthly for 6 months. Data on QoL were collected using the self-completed EQ-5D-5L[31] questionnaire where participants had the capacity to do this. Where a resident could not complete the EQ-5D-5L questionnaire, the proxy version was completed by the carer or a member of staff. We

also collected data on carer QoL using the EQ-5D-5L questionnaire. All QoL data were collected pre-intervention and at 6 months.

## Sample size

This was a pilot study so no sample size calculation was conducted. The number of NHs was chosen on pragmatic grounds to allow testing of study procedures and variability in settings.

## Randomisation

NHs were randomised prior to intervention: four in West Yorkshire and three in Greater London (seven total) to the intervention and four in Yorkshire and three in Greater London (seven total) to 'usual care', stratified by location. We used the SAS version 9.4 statistical program to generate a randomisation list drawn up by a clinical trials unit statistician not involved in the study.

## Blinding

This was not feasible for staff collecting data. Statisticians and health economists were blinded to allocation. The randomisation variable was supplied to them unlabelled, and the main analysis was completed using this.

## Data management

Data were entered onto paper case report forms and then into an encrypted password-protected database in accordance with the UK Data Protection Act and General Data Protection Regulation. We followed a standardised process for database lock.

## Statistical methods

As a pilot study, analyses are mainly descriptive (counts, means, SD, medians with IQR) focusing on recruitment, participant characteristics, other baseline and outcome variables, loss to follow-up and tabulation of SAEs. We summarised completeness of data on outcome measures and questionnaire response rates.

## Economic evaluation

We calculated costs associated with the intervention, including costs of staff training and implementation. Resources use associated with hospital admissions, primary care, and National Health Service (NHS) and social care services were collected using the CSRI and costed from the NHS/personal social services perspective.[32–34] All costs are reported in 2016/2017 British pounds (£). Quality-adjusted life years (QALYs) for residents were calculated using the EQ-5D-5L questionnaire and associated algorithms mapping the 5L descriptive system data onto the 3L valuation set as recommended by the National Institute for Health and Care Excellence.[35–38] We report only results based on QALYs calculated using the resident self-completed EQ-5D-5L questionnaire as these had higher return. There was no discounting of costs or QALYs given they were reported over 6 months only. For missing data we used multivariate imputation by chained equation, generating 20 imputed data sets.

For each of these, we ran 1000 bootstrap replications using non-parametric bootstrapping. Bootstrap results were combined to calculate the mean values for costs and utilities and the SEs around the imputed values used to calculate 95% CI around point estimates. To report the probability that the intervention is cost-effective compared with treatment as usual (TAU) for a range of values of willingness to pay (WTP) for a QALY gained, the bootstrap results have been used to generate a cost-effectiveness acceptability curve[39] and the probability that the intervention is cost-effective compared with TAU at a £20 000 for a QALY gained reported.

## RESULTS

Here, we give trial results with full data given in the supplemental appendices to ensure transparency of reporting. In-depth evaluation of implementation will be published separately.

## Recruitment
### Nursing homes

We recruited the target number of 14 NHs and randomised as planned: seven intervention (three London, four Yorkshire) and seven control (three London and four Yorkshire). One Yorkshire intervention NH was closed by its owners and dropped out pre-intervention. A further intervention NH in London dropped out following PDC training as they were unable to implement the intervention. Most NHs were privately managed with a median 50 residents (IQR 34–68). The majority (73%) were 'dementia registered'. In terms of Care Quality Commission (CQC) ratings, 1 home (7%) was 'outstanding', 11 (79%) 'good' and 2 (14%) 'required improvement' (online supplemental appendix 1).

### Residents

We recruited 237 residents (figure 1), two-thirds were women, predominantly white (90%) with a median age of 86 years (IQR 80–91).

The median Barthel Index score was low at 27 (IQR 9–64) indicating a high level of dependency in activities of daily living. Only 6 residents (3%) had an admission for an ACSC in the pre-intervention period, 11 (5%) residents had at least one ambulance called and 12 (5%) had an unscheduled GP visit or telephone contact (table 2).

### Family carers

We recruited 91 family carers, two-thirds were women (table 2, online supplemental appendix 2). Median age was 63 years (IQR 57–71). Most (91%) wished to be involved in noticing early changes in the resident's health.

### Staff

We recruited 132 staff (online supplemental appendix 3), with a median age of 42 years (IQR 30–53), 12% were men, 50% of nurses spoke English as a first language and 59% of staff described themselves as white. Most staff (71%) had worked at the NH for a year or more and

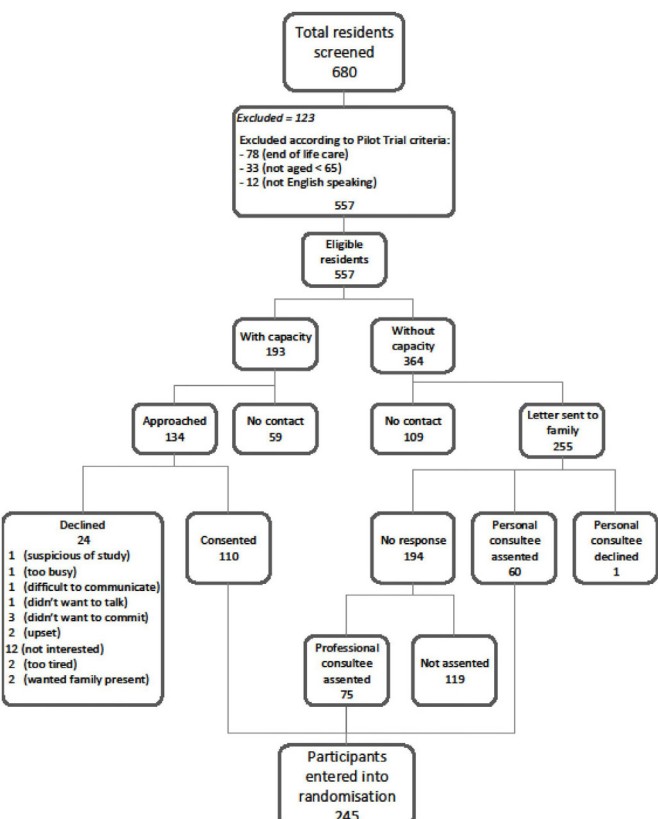

**Figure 1** Resident recruitment flowchart.

30% were qualified nurses. Scores on the P-CAT scale (possible range 13–65) were generally positive with a median score of 49 (IQR 46–53) (online supplemental appendix 4). Most nurses were positive about the quality of communications they had with GPs (online supplemental appendix 5) and their self-rated knowledge and skills (online supplemental appendix 6).

### Primary objective: feasibility and acceptability
#### Use of the intervention
Across the 5 intervention NHs, only 16 Stop and Watch forms were completed of which 11 came from a single NH. Eight care pathways were reported as completed but only three were located by the study team. There was a median of one Stop and Watch form (IQR 0–3) and a median of zero care pathways (IQR 0–2) completed per month. In a few cases, routine clinical observations (eg, temperature, blood pressure and so on) were carried out, but not reported systematically or presented as part of a coherent assessment plan. One home had a policy of recording routine observations once per month.

#### Serious adverse events
There were no differences in SAEs between TAU and intervention groups. There were 104 SAEs in 74 residents during the study and 33 residents died (19 TAU and 14 intervention). Of the 104 SAEs, hospitalisation (any cause) was the most common (N=50) (online supplemental appendix 7).

| Table 2 Characteristics of residents and family carers | | | |
|---|---|---|---|
| | **Cohort** | **TAU** | **BHiRCH-NH** |
| **Characteristic** | **n or median (% or IQR)** | | |
| **Residents** | | | |
| Demographics | N=234 | N=137 | N=97 |
| Male | 73 (31) | 46 (34) | 27 (28) |
| Age | 86 (80–91) | 86 (80–91) | 84 (78–91) |
| Ethnicity | N=225 | N=131 | N=94 |
| White | 203 (90) | 117 (89) | 86 (91) |
| Black | 14 (6) | 9 (7) | 5 (5) |
| Asian | 5 (2) | 4 (3) | 1/(1) |
| Other | 3 (1) | 1 (1) | 2 (2) |
| Marital status | N=223 | N=127 | N=96 |
| Married or cohabiting | 49 (22) | 28 (22) | 21 (22) |
| Single | 59 (26) | 40 (32) | 19 (20) |
| Divorced or widowed | 115 (52) | 59 (46) | 56 (58) |
| Education | N=184 | N=133 | N=71 |
| Completed years of education | 11 (9–12) | 11 (10–12) | 11 (9–11) |
| No qualifications or GCSE or equivalent | 107 (58) | 63 (56) | 44 (62) |
| A Level/NVQ/HNC/HND or equivalent | 18 (10) | 11 (10) | 7 (10) |
| Degree or higher degree | 23 (13) | 14 (12) | 9 (13) |
| Other qualification | 36 (20) | 25 (22) | 11 (15) |
| Function | | | |
| Barthel Index score | 27 (9–64) | 27 (9–66) | 30 (8–63) |
| **Carers** | | | |
| Demographics | N=91 | N=56 | N=35 |
| Male | 31 (34) | 17 (30) | 14 (40) |
| Age | 63 (57–71) | 62 (57–71) | 64 (58–74) |
| Ethnicity | N=87 | N=52 | N=35 |
| White | 72 (83) | 43 (83) | 29 (83) |
| Black | 12 (14) | 7 (13) | 5 (14) |
| Asian | 3 (3) | 2 (4) | 1 (3) |
| Marital status | N=87 | N=53 | N=34 |
| Married or cohabiting | 65 (75) | 36 (68) | 29 (85) |
| Single | 10 (11) | 8 (15) | 2 (6) |
| Divorced or widowed | 12 (14) | 9 (17) | 3 (9) |

Continued

**Table 2** Continued

| Characteristic | Cohort | TAU | BHiRCH-NH |
|---|---|---|---|
| | n or median (% or IQR) | | |
| Education | N=86 | N=53 | N=33 |
| Completed years of education | 11 (11–12) | 12 (11–13) | 11 (11–12) |
| No qualifications or GCSE or equivalent | 35 (41) | 21 (40) | 14 (42) |
| A Level/NVQ/HNC/HND or equivalent | 13 (15) | 9 (17) | 4 (12) |
| Degree or higher degree | 26 (30) | 14 (26) | 12 (36) |
| Other qualification | 12 (14) | 9 (17) | 3 (9) |
| Preferred role | N=87 | N=52 | N=35 |
| Noticing early signs of changes in health | 79 (91) | 49 (94) | 30 (86) |
| Informing staff about early signs of changes in health | 77 (89) | 48 (92) | 29 (83) |
| Educating staff about how early signs of changes in health present | 51 (59) | 28 (54) | 23 (66) |
| Educating care staff about health history of their family member | 57 (66) | 33 (63) | 24 (69) |
| Prefer not to be involved | 5 (6) | 2 (4) | 3 (9) |
| Other | 18 (21) | 9 (17) | 9 (26) |

BHiRCH-NH, Better Health in Residents of Care Homes with Nursing; GCSE, General Certificate of Secondary Education; HNC, Higher National Certificate; HND, Higher National Diploma; NVQ, National Vocational Qualification; TAU, treatment as usual.

## Secondary objectives

1. Resident consent procedures and collection of sufficient individual level data

   We screened 680 residents, 557 met eligibility criteria and 245 were recruited (35% recruitment). Taking into account that two NHs dropped out, leaving 12 in the study, we reached our target of 240 residents (20

residents per NH). Most eligible residents (364, 65%) did not have capacity to consent to participate in the study. Of recruited residents, 73% completed the study (online supplemental appendix 8).

2. Assess intervention fidelity

   It was not possible to assess fidelity to the intervention as the intervention was not implemented as intended and the documentation required to assess fidelity was not available.

3. Assess the effectiveness of the implementation strategy and level of NH staff engagement with the intervention

   The implementation strategy was not effective and NH staff did not engage with or use the intervention tools. Data and learning from implementation will be presented in a separate paper.

4. Investigate whether the intervention would be sustainable outside the context of a trial

   The intervention was not widely used; 16 Stop and Watch forms were completed and we found 8 completed care pathway documents. We concluded the intervention in its current form was not sustainable.

5. Assess potential primary and secondary outcome measures for a definitive trial

   *Rates of hospitalisation for ACSCs*

   At baseline the rates of hospitalisation for ACSCs (respiratory infection, exacerbation of CHF, UTI and dehydration), our potential primary outcome, were low and 0.4% of the cohort had an admission for respiratory infections, 1% for UTI, none for dehydration and 0.4% had an admission for CHF in the month before the trial started. There were six admissions in total from the 235 residents in the study. Considering the whole 6-month study period, 25 study participants (15%) had an unplanned hospital admission for one of the ACSCs. The low rates of unplanned hospital admission (per 100 person months) for these ACSCs suggest this is not an optimal primary outcome measure for future studies table 3). Considering secondary outcome measures for a definitive trial, we found a slightly higher proportion of participants (n=38, 16%) had an A&E attendance during the follow-up period, 42 (18%) had at least one ambulance called, 29 (12%) had an unscheduled (out-of-hours) GP visit and 21 (11%) died. The incidence of these events was still relatively low and not sufficiently frequent to be definitive study outcomes.

   *Hospitalisation for an ACSC as a proxy for avoidable hospital admission*

   We intended to use the SIRR tool to assess the appropriateness of 30 resident admissions for one of the four ACSCs, but it was not always possible to identify the reasons for admission and these were low overall. We therefore expanded our sample to include hospital admissions for any cause. One NH had no eligible residents, because those admitted to hospital had died and/or their care records were no longer available. We therefore also included residents who died in hospital, as long as their records were still available. We were

**Table 3** System-level outcome data

| | Pre-intervention | | | Over 6-month follow-up period | |
|---|---|---|---|---|---|
| | Whole cohort | TAU | BHiRCH- NH | TAU | BHiRCH-NH |
| | n or median (% or (IQR) | | | | |
| **Study cohort** | N=235 | N=139 | N=96 | N=139 | N=96 |
| At least one admission in the last month | 6 (3) | 2 (1) | 4 (4) | 19 (14) | 16/96 |
| Respiratory infection admission | 1 (0.4) | 0 | 1 (1) | 8 (6) | 5/96 |
| Urinary tract infection admission | 2 (1) | 1 (1) | 1 (1) | 5 (4) | 2/96 |
| Dehydration admission | 0 | 0 | 0 | 0 | 1/96 |
| Congestive heart failure admission | 1 (0.4) | 0 | 1 (1) | 0 | 1/96 |
| At least one ambulance called | 11 (5) | 4 (3) | 7 (7) | 20 (14) | 22/96 |
| At least one out-of-hours GP visit or telephone contact | 12 (5) | 11 (8) | 1 (1) | 14 (10) | 15/96 |
| At least one accident and emergency attendance | 10 (4) | 6 (4) | 4 (4) | 17 (12) | 21/96 |
| Died | 1 (0.4) | 1 (1) | 0 | | |
| **Nursing home data** | | | | | |
| Number of hospital admissions | 3 (2–5) | 4 (2–7) | 3 (2–4) | 12 (12–16) | 12 (7–18) |
| Number of ambulances called | 3 (2–6) | 4 (2–9) | 3 (2–6) | 12 (11–17) | 19 (7–22) |
| Unscheduled (out-of-hours) GP visits or telephone contacts | 1 (1–3) | 2 (1–3) | 1 (1–3) | 8 (7–13) | 9 (4–25) |
| Accident and emergency attendances | 3 (2–5) | 3 (1–4) | 3 (2–6) | 12 (11–13) | 8 (7–13) |
| Rate of hospital admissions per 100 person months | | – | – | 5.3 (2.3–8.3) | 5.9 (1.7–7.1) |
| Rate of ambulances called per 100 person months | | – | – | 5.7 (2.3–8.0) | 6.0 (2.0–9.3) |
| Rate of unscheduled (out-of-hours) GP visit or contacts per 100 person months | | – | – | 2.5 (1.8–4.0) | 5.1 (1.1–6.0) |
| Rate of accident and emergency attendances per 100 person months | | – | – | 4.3 (2.5–8.3) | 3.9 (2.0–5.6) |

BHiRCH-NH, Better Health in Residents of Care Homes with Nursing; GP, general practitioner; TAU, treatment as usual.

able to assess 10 admissions from a total of 24, which occurred during the study period. Using the SIRR tool, we deemed 3 of the 10 admissions assessed potentially avoidable. None of the NH care records provided a complete picture of residents' health in the period leading up to admission. This suggested an ACSC admission was not a reliable proxy measure for an 'avoidable admission'.

6. Completeness of data collection and documentation, return rate of questionnaires
   *Care staff-related data*
   We collected data on most recruited care staff at baseline (N=132). For example, 129 care staff gave demographic details (98%) and 127 (96%) completed the P-CAT scale. Attrition in response to these scales was secondary to NHs dropping out of the study, rather than staff being unwilling or unable to complete them.
   *Resident-related data*
   Demographic and functional ability data were available on most residents at baseline (N=235) including

gender (99%), ethnicity (95%), marital status (98%) and Barthel Index (98%). We cannot be sure that no admissions or visits to acute hospital were missed as NHs did not have centralised systems for collecting these data.

7. Cost and outcome data for use in an economic evaluation, key cost components and probability of cost-effectiveness
   One NH withdrew after randomisation and therefore, we considered the cost of intervention to be £0 (as 'per randomised' approach). Assuming the intervention was offered to all 89 residents randomised to the intervention group, the mean cost per resident would be £74 (95% CI £64 to £84). During follow-up, there were no significant differences in the majority of the components of healthcare resource use between the intervention and TAU groups (online supplemental appendices 9 and 10). Differences in mean utility values and QALYs were not statistically significant when carers assessed their QoL and therefore, no further

**Table 4** Cost-effectiveness of BHiRCH-NH intervention versus TAU: complete case and imputed data analyses

|  | Incremental cost | | QALY gained | |
|---|---|---|---|---|
|  | Mean | (95% CI) | Mean | (95% CI) |
| Base case* | £208 | −£561 to £977 | 0.016 | 0.003 to 0.300 |
| Complete case† | £352 | −£745 to £1448 | 0.018 | −0.012 to 0.048 |

*Data include values imputed using multiple imputation with SEs corrected to account for uncertainty in the imputed values. QALYs gained are adjusted for baseline utility values and nursing home clustering. The incremental costs are adjusted for costs in the 1-month period prior to baseline and nursing home clustering.
†As for the base case analysis except there is no multiple imputation for missing data.
BHiRCH-NH, Better Health in Residents of Care Homes with Nursing; QALY, quality-adjusted life year; TAU, treatment as usual.

analysis was performed. The mean total cost of healthcare resource use/resident over 6 months was £1458 (95% CI £1351 to £1566) in the intervention group and £1233 (95% CI £1171 to £1295) in the TAU group (online supplemental appendix 11). Non-parametric bootstrapping after multiple imputation produced a mean total cost per resident in the intervention group of £1479 (95% CI £757 to £2200), compared with £1271 (95% CI £975 to £1566) in the TAU group. The mean difference in cost between the BHiRCH-NH and TAU group was £208 (95% CI −£561 to £977). Non-parametric bootstrapping after multiple imputation produced 0.315 (95% CI 0.304 to 0.326) QALYs in the intervention group and 0.298 (95% CI 0.290 to 0.307) QALYs in the TAU group, generating a statistically significant mean difference in QALYs of 0.016 (95% CI 0.003 to 0.300) (table 4).

The incremental cost per QALY gained of BHiRCH-NH versus TAU was £12 633. Residents receiving the intervention accrued a non-significantly higher cost and a very small increase in QALYs; the intervention has a 65% probability of being cost-effective at a WTP of £20 000 (online supplemental appendix 12).

## DISCUSSION

Our cluster randomised pilot trial of the BHiRCH-NH intervention in 12 NHs found study processes were effective. We successfully recruited, retained and obtained individual-level data from residents, staff and family carers. Adverse event data did not suggest the intervention caused harm. Our CRP worked with us throughout, advising on study set-up, engaging with homes and potential participants, data collection, and contributing to data analysis and interpretation. However, despite excellent recruitment and retention, there was limited engagement with the intervention tools and the support offered for their implementation. The lack of use of the intervention coupled with the economic analysis means that we would not recommend a definitive randomised controlled trial of the BHiRCH-NH intervention. We focused on four key ACSCs (dehydration, respiratory and UTIs, and exacerbation of CHF) because they are common causes of potentially avoidable hospital admissions and a significant area of health policy focus. There is lack of national comparative data, but we found lower hospitalisation rates for these conditions than we expected. Categorising admissions to hospital is complex and people may present with broader symptoms that is, delirium or falls. ACSCs may be more of an administrative label to be used in large data analyses rather than a sensitive tool for assessing the 'avoidability' or otherwise of acute hospital admissions.

Implementing practice change in NHs can be challenging. Despite offering monthly telephone support calls during the study at times agreed with PDCs, they often could not speak due to changes in duty rosters, last-minute leave or being too busy. While the INTERACT programme, from which our BHiRCH-NH intervention was adapted, demonstrated reductions in all-cause hospitalisations among actively participating NHs, a subsequent larger randomised controlled implementation trial in 85 US NHs had no effect on emergency department visits or hospital admissions.[14] Resources, competing demands and instability of NH leadership were all barriers to successful INTERACT implementation,[40] and it is likely that these factors were also present in our study.

Our study has limitations. We recruited NHs, residents, staff and family carers to target, although two NHs dropped out from the intervention arm. Our NHs may be atypical as 79% were rated as 'good' by the CQC, compared with the UK national average of 73%. Residents in our sample were broadly representative of the UK NH population in terms of age and gender.[41] Monthly visits from research fieldworkers and the appointment of research facilitators who were existing NH staff fostered positive relationships between NHs and the research team and facilitated access to NH records. We cannot verify that data on hospital attendances, staffing and support from external healthcare services are complete, as most NHs did not routinely record hospital admissions or ambulance and GP callouts. Staff self-rated their knowledge and skills regarding health conditions and their communication with primary care and research assistants collecting outcome data were not blind to NH allocation. We assumed we could use admission for ACSCs as a proxy for avoidable admissions, but even after using the SIRR tool,[42] it was difficult to identify whether an admission was for an ACSC or not. Future studies should include consent to access hospital records so this can be more thoroughly assessed.

Current UK policy focus on reducing hospitalisations and enhanced healthcare in NHs has led to significant levels of activity in local health and social care services. Research in such a fast-moving landscape is challenging. NHS England's demonstration projects 'Vanguards' and increasing numbers of local quality improvement initiatives means 'usual care' will be improving, making trials challenging to conduct. Although our intervention was

co-designed with staff, this involved an extra burden of observation and documentation, which was difficult for staff who were already pressed for time. Recent studies conducted in care homes suggest that a higher level of support and facilitation may be required for effective implementation; meetings involving all NH staff may be more successful.[43] The Palliative Care for Older People Steps to Success Programme conducted in 78 NHs across seven countries, which involved a complex intervention to improve end-of-life care, did not improve resident comfort or staff knowledge despite using a train-the-trainer approach.[44] However, the Well-Being and Health for People with Dementia programme was successfully implemented in 69 UK care homes and showed a significant effect on reducing agitation possibly through a much higher intensity of external facilitation that may not be sustainable or cost-effective.[45] Since we designed our intervention and conducted our pilot trial, knowledge on improving healthcare for NH residents has increased. Key components for resident healthcare, identified in the 2017 'Optimal' study were GP involvement supported by integrated external services.[46] These findings are reflected by emerging evidence that change to the wider health and social care system may be more effective in reducing the number of NH residents admitted to hospital.[47]

## CONCLUSIONS

Our co-designed complex intervention for early detection, monitoring and communication of change in residents' health does not warrant a future definitive trial because it was not implemented in practice. It was important that this pilot trial was conducted as this has avoided the risk of embarking on a full trial and subsequent waste of resources and time.[48] Our study contributes to learning in the under-researched but vital field of NH research.

**Author affiliations**
[1]Marie Curie Palliative Care Research Department, Division of Psychiatry, University College, London, UK
[2]Barnet Enfield and Haringey Mental Health Trust Liaison Psychiatry Team, North Middlesex University Hospital, London, UK
[3]Centre for Applied Dementia Studies, University of Bradford, Bradford, UK
[4]International Observatory on End of Life Care, Lancaster University, Lancaster, UK
[5]Department of Primary Care and Population Health and Priment Clinical Trials Unit, University College London, London, UK
[6]Divisions of Nursing, Occupational Therapy & Arts Therapies, School of Health Sciences, Queen Margaret University, Edinburgh, UK
[7]School of Pharmacy and Medical Sciences, University of Bradford, Bradford, UK
[8]Newcastle University Institute for Ageing and Institute for Health & Society, Newcastle University, Newcastle upon Tyne, UK
[9]Academic Unit of Elderly Care and Rehabilitation, University of Leeds, Bradford Institute for Health Research, Bradford, UK

**Acknowledgements** The authors would like to thank the members of our carer reference panels and the Alzheimer's Society research network volunteers. We are grateful to the nursing homes that supported our feasibility study. We are grateful to the Clinical Research Network staff: Mary-Jo Doyle, Deborah Cooper, Kim Williams, Sally Gordon and Helen Permain, and others who assisted with data collection: Justin Chan, Ayesha Dar, Luiza Grycuk, Dr Adam Hughes, Dr Maria Ivanov and Angela Richardson. We would like to thank the study steering group (Dr Najma Siddiqi (Chair), Professor Tom Dening, Bert Green, Mike Bradburn, Laura Mandefield and members of the international advisory group (Professor Finbarr Martin (Chair), Professor Barbara Bowers, Professor John Gladman, Professor Claire Goodman, Professor Martin Green, Professor Raymond Koopmans, Professor Julienne Meyer, Professor Des O'Neill, Professor Jo Ouslander, Dr Hilary Rhoden, Jenny Adams, Professor Lynn Chenoweth, Professor Graham Stokes, Louise Taylor, Gavin Terry, Steve Williams).

**Contributors** ELS, AF, AB, KF, RH, LM, BM, SN, CP, MP, GR, LR, BW-C, JY and MD were involved in the development of the intervention and made substantial contributions to the concept and design of the study and protocol. LM, MP and RH analysed the data. ELS and AF wrote the paper with assistance from CP, AF, AB, KF, RH, LM, BM, SN, MP, GR, LR, BW-C, JY and MD who critically revised the manuscript and approved the final version.

**Funding** This work was supported by the UK NIHR grant number RP-PG-0612-20010.

**Disclaimer** The views expressed are those of the author(s) and not necessarily those of the NHS, the NIHR or the Department of Health and Social Care. The funding body had no role in the design of the study, collection, analysis and interpretation of data, and in writing the paper.

**Competing interests** None declared.

**Patient consent for publication** Not required.

**Ethics approval** Ethical approval was given by the Queen Square London Research Ethics Committee (reference 17/LO/1542).

**Provenance and peer review** Not commissioned; externally peer reviewed.

**Data availability statement** Data are available upon reasonable request. The data sets generated and/or analysed during the current study are not publicly available, the small number of nursing homes means that data may be identifiable. Data are available from the corresponding author on reasonable request.

**ORCID iDs**
Elizabeth L Sampson http://orcid.org/0000-0001-8929-7362
Rachael Hunter http://orcid.org/0000-0002-7447-8934
Catherine Powell http://orcid.org/0000-0001-7590-0247
Louise Robinson http://orcid.org/0000-0003-0209-2503

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
