## [Reviewer comments · BMJ Open]

ARTICLE DETAILS

TITLE (PROVISIONAL)	A pilot cluster randomised trial of an evidence-based intervention to reduce avoidable hospital admissions in nursing home residents (Better Health in Residents of Care Homes with Nursing - BHiRCH-NH study)
AUTHORS	Sampson, Elizabeth; Feast, Alexandra; Blighe, Alan; Froggatt, Katherine; Hunter, Rachael; Marston, Louise; McCormack, Brendan; Nurock, Shirley; Panca, Monica; Powell, Catherine; Rait, Greta; Robinson, Louise; Woodward-Carlton, Barbara; Young, John; Downs, Murna

VERSION 1 – REVIEW

REVIEWER	A/Prof Magnolia Cardona Institute for Evidence-Based Healthcare, Bond University, Gold Coast, Australia
REVIEW RETURNED	21-Jun-2020

GENERAL COMMENTS	Thank you for the opportunity to review this complex and resource-intensive multicomponent intervention. I am in awe of the extent of recruitment and follow up since this was only a pilot. I commend the authors in their incredible attempt to test the feasibility, level of staff engagement, acceptability and sustainability of the processes and outcomes before deciding whether to embark on a full-blown study. The assumptions and staff involved in measurements of avoidability of admission for ACSC were appropriate. The use of proxy quality of life versions of questionnaire when residents were unable to complete, was also very acceptable practice. The authors provided a project handbook, delivered a workshop on 4 ambulatory care sensitive conditions for NH staff, and offered monthly telephone coaching to the local champions in NH. Another plus of the design was the involvement of local champions and consumer representatives throughout. While the study sites were a convenience sample including a variety of sizes and locations to enhance generalisability, assignment to intervention or control arms was conducted at random by an external party. The authors acknowledged the infeasibility of blinding the intervention group and deviations from protocol in terms of trial duration for reasons beyond their control. My suggested minor revisions below are typos and optional enhancements. Page 9, lines 20-22, Page 10, lines 6-7 and page 14, lines 14-16: the text announces four ACSCs but only mentions three of them. Pneumonia missing from all.
--

	Page 11, lines 5-16: the baseline was collected for the month before the intervention but the post-intervention data was collected for 6 months. Seasonal influences are known to affect the contacts with the health system , so this must be acknowledged in the limitations of the study. The numbers of responses in Appendices 5 and 6 are so small, it would make more sense to present them in descending order of frequency overall for the cohort at the pre-intervention stage to make interpretation more visually meaningful. Appendix 7 could be made more informative and self-explanatory by adding more specific details to the title and specifying the type of adverse events NOT caused by the intervention, for instance. The unfortunate finding that the intervention was not implemented, even though it was “enhanced usual care” highlights the difficulties in uncovering health status in the period leading to admission, documenting information on reasons for hospital transfers, and changing culture in health and supportive services regardless of assistance with implementation. Evaluating health interventions is therefore a challenge, even with the imputation techniques used. It is hard to imagine what else the authors could have done to enhance the local capacity to implement and manage ACSC to prevent hospital transfers, other than select NH with larger number of pre-implementation transfers, provide financial incentives for implementation to the sites (which makes it unsustainable) or make the intervention mandatory (which obviously is not welcome in all health systems). I am not sure I agree with the statement on page 25 (lines 6-10) about the low sensitivity of ACSC. I believe the major hurdle in this multicomponent intervention with clear objectives and comprehensive plan was the choice of NHs with lower prevalence of ACSC (the outstanding and good NHs), and the burden of data collection with implementation tools to achieve detailed documentation of all aspects planned for. These additional requirements in the face of completing clinical demands -as mentioned by the authors on page 24- without time availability or supplementary staff or funding could deter real-world staff participation, and as disappointing as it was, it is good it happened during the pilot to prevent further losses. Alternatively, if they authors had started the pilot during a busy season to find more of the target outcomes and had included linkage to hospital data, the outcome could have been different. It was good to see that suggested in the discussion. It would be useful for readers to see a comment by the authors in the discussion to understand whether not all NH types require this type of intervention. Perhaps a more targeted approach to benefit the “high-risk_ NHs is appropriate. The recommendation for higher level of support for successful implementation (page 25, lines 15-23) may be true but makes enhanced routine care non-sustainable without additional injection of funds (as later noted on page 26, line 35 from another UK example. In my opinion, the authors used prior evidence to design their intervention, provided sufficient detail to enable replicability, but also humbly acknowledged the challenges and scarcity of Information to proceed to a full trial. I agree with their conclusion.
--	---

	Releasing lessons learnt and not-so-positive results is an uncommon but welcomed addition to knowledge and debate in health services research. I commend the authors for the effort and transparency to prevent duplication of effort by others when implementation is not viable.
--	--

REVIEWER	Morten Lindbæk Antibiotic centre for primary care Dept of general practice University of Oslo Norway
REVIEW RETURNED	30-Jun-2020

GENERAL COMMENTS	General comments: This is an interesting study as few comprehensive intervention studies have been performed in the nursing home setting. The study had very ambitious plans with a number of data to be reported, and was importantly cluster-randomised. It is important that the study is published although the intervention did not work, as researchers have to learn how to implement intervention in challenging settings. The data collection tools seem to be relevant for the purposes. The clinical conditions to be studied are important and the purpose of avoiding unnecessary hospital admissions for frail elderly people is of great significance. The study managed to recruit an adequate number of residents for the study. They achieved to register important data on hospital admissions, which is important as a baseline for later studies. I have some specific comments to be considered before publishing:  1. It was known for the researchers that the nursing home setting is challenging, with high turn-over of personnel, large proportion of staff without nursing education. Thus the implementation support plan seems to be too optimistic. In the support plan it is stated that the PDCs should establish a working group at each NH. How was this stimulated and followed up? This was a key point in the implementation. Did you consider a more comprehensive plan on this point, for example make a large implementation meeting at each NH where all personnel were supposed to attain? I understand that the evaluation of the implementation is coming in a new paper, but I suggest to take up some of these points also in the current paper. When a pilot study is made it is crucial to maximize the chance of success to find what is feasible and realistic in a larger study. 2. How was the implementation followed up during the 6 months? What were the back up plans when things were not working? Under support it is stated that you offered telephone coaching monthly. Why was this not compulsory for all NH? I miss data on this point. By more close follow up it would have been possible to catch how the implementation was working and in this case not working in most NHs. Then you would have the chance to put in extra resources to make the intervention work. 3. Under methods it is stated that patient and public was involved in the study. However, I miss whether personnel in relevant NHs were engaged in planning of the protocol and the implementation. This would have increased the chance of success and how to meet obstacles in the study. 4. I miss reference to our own intervention in Norwegian NH in 2009-2011 (Romoren et al). This was a comprehensive
--

	intervention on enhancing use of iv fluid and antibiotics with the same aim as in this study: to avoid unnecessary hospital admissions. We used a stepped wedge design in over 30 NHs, and found a significant fall in admissions. In this study we had implementation meetings with all personnel in each NH, which we think was crucial to make it successful. 5. I miss a more detailed description of the organization of the nursing homes and an evaluation of the differences between NHs in Europe and in US. This is important for the external validity of the study. Ref: Romøren M, Gjelstad S, Lindbæk M. A structured training program for health workers in intravenous treatment with fluids and antibiotics in nursing homes: A modified stepped-wedge cluster-randomised trial to reduce hospital admissions. PLoS One. 2017 Sep 7;12(9):e0182619. doi: 10.1371/journal.pone.0182619. eCollection 2017.
--	--

VERSION 1 – AUTHOR RESPONSE

REVIEWER 1

Page 9, lines 20-22, Page 10, lines 6-7 and page 14, lines 14-16: the text announces four ACSCs but only mentions three of them. Pneumonia missing from all.

Response: this has been corrected to consistently include “respiratory infections”

Page 11, lines 5-16: the baseline was collected for the month before the intervention but the post-intervention data was collected for 6 months. Seasonal influences are known to affect the contacts with the health system, so this must be acknowledged in the limitations of the study.

Response: In the methods we state that the pilot trial ran for 10 months (November 2017-August 2018) thus it covered both Summer and Winter seasons where there is greatest variation in admission rates in the UK.

The numbers of responses in Appendices 5 and 6 are so small, it would make more sense to present them in descending order of frequency overall for the cohort at the pre-intervention stage to make interpretation more visually meaningful.

Response: We agree that numbers are small but have decided to leave the table as –is so that the questionnaire items run in the order they are presented to participants.

Appendix 7 could be made more informative and self-explanatory by adding more specific details to the title and specifying the type of adverse events NOT caused by the intervention, for instance.

Response: We have presented this as per standard UK practice in trials which is to list all potential SAEs, even those which aren’t clearly attributable to the intervention. We also include further detail in the table in rows “Expected side effect or outcome of the intervention” and “Not expected”.

It would be useful for readers to see a comment by the authors in the discussion to understand whether not all NH types require this type of intervention. Perhaps a more targeted approach to benefit the “high-risk_ NHs is appropriate.

Response: We have added a comment regarding this into the discussion (paragraph 1).

REVIEWER 2

Thank you for your comments and suggestions, particularly regarding the implementation of our intervention and noting that a full paper will be published describing this process in far more detail.

We have attempted to add some further detail as suggested by your comments, but within the constraints of the journal word count:

The implementation support plan seems to be too optimistic. In the support plan it is stated that the PDCs should establish a working group at each NH. How was this stimulated and followed up? This was a key point in the implementation. Did you consider a more comprehensive plan on this point, for example make a large implementation meeting at each NH where all personnel were supposed to attend? I understand that the evaluation of the implementation is coming in a new paper, but I suggest to take up some of these points also in the current paper.

Response: We have added more detail regarding our rationale for the implementation plan and the PARHIS framework used to inform this in the methods section (subheading "Implementation support")

How was the implementation followed up during the 6 months? What were the back up plans when things were not working? Under support it is stated that you offered telephone coaching monthly. Why was this not compulsory for all NH? I miss data on this point. By more close follow up it would have been possible to catch how the implementation was working and in this case not working in most NHs.

Response. More detail on this will be provided in our paper on the implementation process. However, we have given some additional information on the telephone coaching in the discussion (paragraph 2) and the reasons why we think this was not a successful approach.

Under methods it is stated that patient and public was involved in the study. However, I miss whether personnel in relevant NHs were engaged in planning of the protocol and the implementation.

Response: staff were integral to intervention development and this is stated in the introduction, paragraph 4.

I miss reference to our own intervention in Norwegian NH in 2009-2011 (Romoren et al). ...In this study we had implementation meetings with all personnel in each NH, which we think was crucial to make it successful.

Response; Thank you for signposting us to your very interesting paper. We have now included the importance of meetings with the whole staff body and have referenced your paper (discussion, paragraph 4, reference number 44)

I miss a more detailed description of the organization of the nursing homes and an evaluation of the differences between NHs in Europe and in US.

Response: we were constrained by the word count in giving a detailed explanation. In addition the UK care home market is very heterogeneous. We have referenced a document which does give this contextual detail (reference 1, Lang and Buisson 2017)

VERSION 2 – REVIEW

REVIEWER	Morten Lindbæk Dept of general practice University of Oslo Norway
REVIEW RETURNED	11-Sep-2020
GENERAL COMMENTS	Response to author's response: This is my evaluation of the response, point by point Referee 1. a) Response ok b) Pilot trial preferable 12 months but response is acceptable. c) Response ok

	d) Table/appendix should be self-explanatory, and author should consider change of title e) Response ok. Referee 2. a) This point could be elaborated somewhat more b) As stated, this is an interesting point, but I accept that this will be a part of the next paper, and look forward to seeing it. c) This is summarized in introduction para 4, but I would like to see more details, also in this paper. d) Response ok e) This is only a general reference, I still miss some more details to describe UK nursing homes compared with other European NH and in the US.
--	--

VERSION 2 – AUTHOR RESPONSE

We would like to thank the reviewers again for the time and care they took in reading our paper and their very helpful and constructive comments. We have attempted to respond to these, without going too far over the journal word count which is now at 5285.

Our amendments have been highlighted in red text.

REVIEWER 1

Comment: Appendix 5 (point “d” in email sent from the journal) “Table/appendix should be self-explanatory, and author should consider change of title”

Response: We have modified the title of the appendix table to “Serious adverse events categorised by type and potential relationship to the intervention”. We have also added the Standard definition, used in UK research, as mandated by the UK Health Research Authority, as a footnote to the table. We have added more information into the appendix table to enhance understanding.

REVIEWER 2

Thank you for your comments and suggestions particularly regarding the implementation of our intervention and noting that a full paper will be published describing this process in far more detail.

Comment: “This point could be elaborated somewhat more” ...(from original set of review comments...)”The implementation support plan seems to be too optimistic. In the support plan it is stated that the PDCs should establish a working group at each NH. How was this stimulated and followed up? This was a key point in the implementation. Did you consider a more comprehensive plan on this point, for example make a large implementation meeting at each NH where all personnel were supposed to attend? I understand that the evaluation of the implementation is coming in a new paper, but I suggest to take up some of these points also in the current paper”.)

Response: We have added more detail as requested, but within the constraints of the journal word count. Under the subheading “Implementation support” on page 8 we have:

- Added details of the introductory meeting
- Given more information on the project handbook
- Added more detail about the regular support and monitoring of the intervention provided by the research team

Comment: “This is summarised in introduction para 4, but I would like to see more details, also in this

paper”...(from original set of review comments “However, I miss whether personnel in relevant NHs were engaged in planning of the protocol and the implementation”)

Response: We have added more detail: “This included 18 semi-structured interviews and three consensus co-design workshops over five months, with amendments made to the intervention after each workshop. Participants comprised 22 diverse stakeholders (two care home managers, three care assistants, eight nurses, four general practitioners, three family carers, a geriatrician and quality improvement manager) (paper in preparation).” Page 5 paragraph 1

Comment: “This is only a general reference, I still miss some more details to describe UK nursing homes compared with other European NH and in the US”

Response: It is very challenging within the word count limits of this journal to provide detailed information on differences between UK, European and US nursing homes. This is because there is great heterogeneity, even between European countries in how care homes are funded, staffed and regulated. The American care home market is diverse and complex. We have added more detail on UK nursing home, how they are funded, and, of particular relevance to this paper, how medical care is provided:

“Most UK NHs are owned by private companies and residents are expected to pay on a means-tested basis. Unlike some European NHs, for example in the Netherlands, there is no on-site provision of medical care. NH residents are served by a general practitioner and visiting staff such as specialist nurses. Older people living in NHs have complex healthcare needs with high levels of multi-morbidity, frailty and dementia. The King’s Fund² and British Geriatrics Society³ have raised concerns about the inconsistency and quality of healthcare provision to NHs.” Page 1 paragraph 1.

We hope that we have now adequately responded to the comments and that the paper will be accepted for publication.

VERSION 3 – REVIEW

REVIEWER	Morten Lindbæk Antibiotic centre for primary care Dept of general practice University of Oslo, Norway
REVIEW RETURNED	21-Nov-2020
GENERAL COMMENTS	I have gone through the authors’ response to the last comments of the reviewers. There are some limitations that I accept; the word count of BMJ Open and some details left for the next paper on the study. Based on this I suggest accept.